# Bone Union Quality after Fracture Fixation of Mandibular Head with Compression Magnesium Screws

**DOI:** 10.3390/ma15062230

**Published:** 2022-03-17

**Authors:** Marcin Kozakiewicz, Izabela Gabryelczak

**Affiliations:** Department of Maxillofacial Surgery, Medical University of Lodz, 113 Żeromskiego Str., 90-549 Lodz, Poland; gabryelczakizabela@gmail.com

**Keywords:** magnesium, mandible head, mandible condyle, condylar head fracture, fracture treatment, bone union, fixing material, osteosynthesis, open rigid internal fixation, surgical treatment

## Abstract

For some years now, fixation devices created with resorbable magnesium alloys for the mandibular head have been clinically available and are beginning to be used. It is thus valuable to evaluate the quality of unions in these cases. The aim of this study was radiological comparison of magnesium versus titanium open reduction and rigid fixations in the mandible condylar head. Thirty-one patients were treated for fractures of the mandibular head with magnesium WE43 alloy headless compression screws (diameter 2.3 mm) and, as a reference group, 29 patients were included with similar construction titanium screws (diameter 1.8 mm). The 12-month results of the treatment were evaluated by the texture analysis of CT. Near similar treatment results were found with magnesium screws in traditional titanium fixation. Magnesium screws result in a higher density of the bone structure in the mandibular head. Conclusions: The quantitative evaluation of bone union after surgical treatment of mandibular head fracture with magnesium compression headless screws indicates that stable consolidation was achieved. Undoubtedly, the resorption process of the screws was found to be incomplete after 12 months, evidenced by a marked densification of the bone structure at the fracture site.

## 1. Introduction

Fractures of the condylar process in the mandible represent up to half of all mandibular fractures [1,2,3]. The incidence of mandibular head fractures appears to have been underestimated. Since the introduction of the now routinely-performed computerized tomography method, the number of diagnosed head fractures has increased. However, there is a second factor limiting the known number of these fractures—the high complexity of the surgical procedure required for reduction and fixation. Centers that do not have these treatment options make no effort to recognize this pathology. They only focus on functional therapy using two weeks of maxillo–mandibular fixation, guiding elastics, occlusion control, and physiotherapy. Many centers do not utilize this treatment approach due to the fear of facial nerve palsy and atrophy of the proximal fragment of the mandible head. The potential long-term complications of a non-surgical route, such as temporomandibular joint disorder or craniomandibular dysfunction, osteoarthrosis, pain, or ankylosis, are concerns the therapeutic team is aware of and accept [4].

However, open rigid internal fixation (ORIF) provides restoration of the original mandibular head anatomy, restoration of the physiological position of the articular disc, natural joint function, and a short healing process, compared to closed conservative treatment. Surgical treatment of mandibular head fractures is becoming increasingly common with the use of long screw fixation through the lateral fragment end of the ascending ramus [5]. Due to anatomical difficulties in using more screws [6,7] and technical challenges, a specialized set is required—low profile screws [8,9,10]. This solves many clinical issues and prevents iatrogenic destruction of mandibular head fragments by thick screws, as well as serious functional complications [11]. Unfortunately, adequate reduction is an ongoing problem, and osteosynthesis remains under considerable stress in the masticatory system. 

This rigid fixation of reduced fractures in the mandibular head can be achieved by using three types of materials: non-resorbable titanium screws [8,12,13,14,15,16], resorbable polymer screws [8,14,15,16], and resorbable magnesium screws [17,18].

The overwhelming majority of open rigid internal fixation (ORIF) procedures of bones are currently done with titanium materials. This is true for all parts of the skeleton, including the mandibular head. This is due to the accustomed stiffness characteristic of titanium alloys, but it is not desirable for such fixations to remain inside the human body after a period of complete fracture consolidation. An increase in free radical formation or the development of inflammation in the tissues surrounding osteosynthesis materials in the mandible has been documented for years. Moreover, the exposure of titanium alloys outside the bone and in contact with soft tissues induces apoptosis in the periosteum [19]. Titanium materials cause oxidative and nitrosative stress in cells and significantly disturb mitochondrial function [20,21]. These are the reasons why screws should be removed from the bone after union is achieved. However, reoperation in the mandibular head is not an easy or simple matter—it is associated with the risk of facial nerve palsy. The introduction of resorbable polymers seems to solve this problem, however, only partially. Early monomeric forms of biodegradable implants have been shown to be associated with delayed degradation (>5 years), which in itself is not negative. However, during degradation, these polymers cause a foreign-body reaction, fistulas, osteolytic pathological lesions, and a variable size of swellings around the fixation area [22,23,24]. As developments in creating copolymers, self-reinforcing materials, and modulating the degradation process progressed [25,26,27,28], interesting and good resorbable screws began to appear. The main disadvantage remained—low mechanical strength [29]. This is because polymer screws are usually thicker than titanium screws in the case of mandibular head osteosynthesis [30]. The use of polymer screws requires a reduction in bone fragments, provisional stabilization, drilling of both fragments, tapping the hole, and finally screwing together with a thick screw with limited rigidity. This is too long a series of manual steps to be performed in the mandibular head fixation procedure, which is difficult and challenging. A little help is provided by resorbable ultrasonically melted pins [14]. Therefore, resorbable magnesium osteosynthesis, combining both the spontaneous disappearance of the screw after completion of the consolidation period (similar to polymers) with the mechanical strength of the metal, should be viewed as a third way to achieve successful treatment and fracture of the mandibular head.

Resorbable materials [31,32,33] are sought after for mandibular head surgery; initial attempts of surgical treatment using resorbable metal (i.e., magnesium alloy) have already been made [34] and a few products are currently available. Magnesium screws have anti-inflammatory properties [35,36], antitumor effects [37,38,39], antibacterial properties [40,41,42], and osteogenesis inductivity [43,44,45]. Moreover, a lower Young’s modulus (near the value of the cortical bone) is an advantage [46]. The main advantage is resorbability and the main disadvantage is the unknown fate of resorption remnants in the human body. To date, there have been no scientific studies on the superiority of magnesium over titanium in mandibular head osteosynthesis. This is probably due to the considerable technical difficulties in treating these fractures. Only a few research centers are addressing this clinical issue. Such a center would need to have an adequate group of patients treated with the standard material (i.e., titanium) to form a reference group. Another factor is probably the novelty of introducing a resorbable metal material. This is followed by the very small number of manufacturers supplying screws, high price of the product, and low availability to research centers.

Due to its mechanical properties, a magnesium system of 2.2 screws seems proper and system 1.7 can be recommended for multi-screw fixations [17,47]. This problem is new and it seems necessary to investigate the quality of consolidation of mandibular head fragments. Such a study has already been mentioned [18].

The aim of this study is radiological evaluation of the quality of bone union after osteosynthesis of a mandibular head fracture with magnesium compression screws.

## 2. Materials and Methods

Sixty-one patients with mandibular head fractures were included in the study. They were divided almost equally into patients treated with magnesium screws (32 patients) and those treated with titanium screws (29 patients who constituted the reference group). The inclusion criteria were: early posttraumatic period, diagnosis of either type B or type C mandibular head fracture [48], applied surgical treatment, preauricular surgical approach to the fracture, and complete radiographic records. Patients were randomized: Tuesday and Thursday patients were operated on with titanium screws, while Wednesday and Friday patients received magnesium fixations. The exclusion criteria for the study were: diagnosis of mandibular head fracture type A [48], absence of the patient at the follow-up examination, closed treatment i.e., conservative treatment, long post-traumatic period i.e., more than 4 weeks after the injury. In the presented study, dedicated mandibular head osteosynthesis compressive screws manufactured by ChM (ChM, Juchnowiec Kościelny, Poland) were used [17].

The demographic information of patients included in this study are presented in Table 1.

Magnesium 2.2 mm headless compression screws [10] (test group) were used for the 32 patients. Magnesium alloy was: MgYREZr (i.e., WE43). No cannulated screws were used in the study, only solid ones (Figure 1 and Figure 2). Titanium headless compression screws (1.7 mm) were used in this study in the 29 cases in the reference group. The titanium alloy was: Ti_6_Al_7_Nb (Table 2 and Table 3). All patients were operated under general anaesthesia with nasotracheal intubation. Maxillo–mandibular ligation was not done during the surgery. After manual reduction and anatomical positioning of the bone fragments, the pilot canal was drilled with a narrow drill and a screw was inserted. This procedure is done partly due to the fact that screws are self-tapping and self-drilling. In addition, there are limitations to the force that can be applied during osteosynthesis. Too much force (used when driving a self-drilling screw) can cause displacement of bone fragments.

Radiographic texture was used as a tool to assess the quality of bone union in this study. All patients included in the study underwent a spiral multislice computer tomography of the mandible 12 months after bone fixation surgery. Bone window was used for data acquisition (window level: 300 and window width: 1500 HU) in RadiAnt viewer (Medixant, Poznan, Poland). The region of interest included the fracture line, excluding the screw images (ROI Post-Fracture). The rest of the mandibular head cancellous bone was the control region (ROI Control). These region of interests (ROIs) were normalized (*μ ± 3σ*) to share the same average (*μ*) and standard deviation (*σ*) of optical density within the ROI. The selected image texture feature (sum of squares) in ROIs were calculated for the post-fracture site and control cancellous bone in the condylar head (Figure 3) [18]:(1)$$\mathrm{SumOfSqrs}=\sum_{i=1}^{N_{g}}\sum_{j=1}^{N_{g}}{\left(i-\mu_{x}\right)}^{2}\;p\left(i,j\right)$$
where Σ is the sum; *μ_x_* is the mean of the row sums of the co-occurrence matrix; *N_g_* is the number of distinct grey levels in the quantized CT image; *p*(*i,j*) is the number of times there is a run of length and *j* having grey level *i* are optical density of pixels two-image-point distant one from another.

Subsequent image texture features (difference entropy from the co-occurrence matrix, and long-run emphasis moment from the run-length matrix) in ROIs were calculated for the control bone and the post-fracture site as:(2)$$\mathrm{DifEntr}=-\sum_{i=1}^{Ng}p_{x-y}\left(i\right)log\left(p_{x-y}\left(i\right)\right)$$
where *p* is probability, *log* is common logarithm [49] and:(3)$$\mathrm{LngREmph}=\frac{\sum_{i=1}^{Ng}\sum_{k=1}^{Nr}k^{2}p\left(i,k\right)}{\sum_{i=1}^{Ng}\sum_{k=1}^{Nr}p\left(i,k\right)}$$
where *Nr* is the number of series of pixels with density level *i* and length *k*; *Ng*—number of levels for image optical density; *Nr*—number of pixel in series [50,51]. These two equations were used for the Bone Index (BI) construction [52], representing the ratio of the measure of the diversity of the structure observed in the radiograph to the measure of the presence of uniform longitudinal structures:(4)$$\mathrm{Bone}\;\mathrm{Index}=\frac{\mathrm{DifEntr}}{\mathrm{LngREmph}}=\frac{(-\sum_{i=1}^{Ng}p_{x-y}\left(i\right)log\left(p_{x-y}\left(i\right)\right))\sum_{i=1}^{Ng}\sum_{k=1}^{Nr}p\left(i,k\right)}{\sum_{i=1}^{Ng}\sum_{k=1}^{Nr}k^{2}p\left(i,k\right)}$$

Texture of bone images were analyses in MaZda 4.6 software developed by the University of Technology in Lodz, Poland [53].

The results of SumOfSqs and BI were related to the level of these features in the control regions to make the result independent of the value of the texture feature in the control. This was done by dividing the value for the Post-Fracture ROI by the value calculated for the Control ROI. A result greater than 1 indicates that the SumOfSqs have a higher value at the fracture line location than in the control region. On the other hand, if the “Relative” value is less than 1, it indicates that the BI value is lower at the fracture site than the control value.

Mathematical results of the image analysis were evaluated using *t*-test and W-test according to the observed distribution (Statgraphics-StatPoint Technologies, Inc., The Plains, VA, USA).

## 3. Results

LngREmph was uniformly elevated at union sites, compared to the control cancellous bone. There was no difference between the 2.14 ± 1.10 in the test group and 1.78 ± 0.61 in the reference group, as measured at the post-fracture site. The cancellous bone image of the control had a significantly lower value for this texture feature (control for the test group 1.35 ± 0.19 and for the reference group 1.23 ± 0.05, respectively). DifEntr evaluation at the union site with magnesium material indicates the same chaotic texture pattern as in the control cancellous bone (1.25 ± 0.13 vs. 1.28 ± 0.09, respectively). In contrast, the union site in the reference patients had a significantly less chaotic i.e., lower entropy texture pattern (post-fracture site 1.28 ± 0.06 vs. control 1.32 ± 0.04).

Through CT, it was found that union of the mandibular head fragments was routinely achieved. The structure (SumOfSqr) of the bone at the union site was significantly different (Table 4) in both the test and reference osteosynthesis groups (it is higher), compared to the cancellous image of the control bone (*p* < 0.05).

## 4. Discussion

Magnesium is a promising material. It is a remarkable composite of mechanical and biomedical properties that has made it suitable for a vast range of medical applications. Many of these inherent properties can be further improved with alloying [54]. For the authors, the most important feature of this metal is its ability to resorb within a living skeleton. In vivo resorption observations [55] revealed that degradation of AZ series (AZ31, AZ91) and rare earth (WE43, LAE442) containing magnesium alloys at the bone-implant interface in guinea pigs completely depended on the alloying elements. New bone in the periosteal and endosteal regions was deposited at the bone implant interface. If it is assumed that healing occurs in the same way in the minipig as in humans [56], then the native bone formation was better using magnesium implants than that of degradable polymer. However, fixing of facial skeletons using biodegradable implants is seldom researched. In one study, a new type of hollow screw made of WE43 magnesium alloy was inserted in the mandible of a minipig [57]. It confirmed bone deposition over the screw and that six months is too limited a period for significant screw resorption. The MgYREZr alloy and titanium implants have also been used for repairing the frontal bone in a minipig. Osteotomy lines in both magnesium and titanium groups have been shown to heal. Lacunas were formed between the implant and the bone. Their formation did not affect the healing process, but their significance is unknown [58]. A rectangular WE43 magnesium implant with and without plasma electrolytic coating was inserted in the nasal bone of a minipig. The coating was successful only in reducing gas formation and increasing Ca–P layer formation on WE43. The bending strength of coated and uncoated WE43 decreased from 92 N before implantation to 86 N after 6 months [59]. Compressive headless screws (uncoated) were tested to show that resorption influence in vitro for axial pull-out force caused a decrease from 399 N to 102 N after 4 months [47]. The plasma coating increased corrosion resistance, improved new bone formation, and reduced gas formation in vivo. The screw was not displaced over six months, as observed with micro CT. All the osteotomies in the orbital rim and the zygoma region were healed without any dislocation of bone or breakage of implant. In addition, the osteotomies performed at the ribs also healed; however, some of the magnesium screws broke due to increased mechanical load during breathing. The degradation rate does not vary significantly in miniature pigs even though WE43 magnesium is implanted in different sites [60,61,62,63]. In contrary, in small animal models such as rats and rabbits, it is reported that the degradation rate of magnesium varies with the implantation site. Huang et al. [64] chose a goat model in which they implanted pure magnesium for treating a fracture in the femur head. The trend of bone remodeling observed over a period of 12 months was similar to that of the human bone. Only 45% of the implant was degraded by the end of 12 months. This indicates the relatively higher resorption resistance of pure magnesium, which might be due to the absence of secondary phases. The currently available scientific literature on maxillofacial surgery is highly lacking in studies on applications of magnesium osteosynthesis in humans.

In this experiment, the evaluation of the bone structure featuring the long term after fixation reveals a slightly elevated bone density with diffuse opaque structures immersed in normal opaque density. Thus, typically, the fracture site changes its structure to a slightly denser one. Bone consolidation at the osteosynthesis site [18] was achieved in the magnesium group as well as in the reference group. In the texture assessment using the co-occurrence matrix (SumOfSqr), a change in bone structure can be seen at the site of the consolidation bone; however, it is worrying to observe a significant difference from the control bone structure. This may be a weakness of the one-parameter texture assessment. Perhaps, the bone feature was not well matched to the biological process being monitored (despite there being noticeable statistical differences between the groups). On the other hand, it can be suspected that the effect of the 12-month screw resorption affects not only the fracture gap, but also the entire mandibular head, which it penetrates thoroughly with the screws. When considering this issue, we looked at the relative difference in texture i.e., in relation to the control bone. It was seen that the relative change in texture characteristics relative to the control image of cancellous bone is the same as in magnesium (both SumOfSqs and Bone Index) as well in in reference osteosyntheses.

The Bone Index is designed to indicate sites with a physiological number and arrangement of bone trabeculae [52]. There is a low BI at post-fracture sites because islands of bone densities (those indicated by the SumOfSqr feature) are more homogeneous (compact) than cancellous bone. This results in a less-chaotic structure, i.e., entropy is reduced (and this entropy is the numerator of the fraction forming the BI). For the same reasons, the bone image here has broad and uniform radio-opague fields, where long lines of pixels of the same optical density can be found. This causes high results for the LngREmph calculation, and this is the number found in the denominator of the fraction that forms BI. Thus, it affects the reduction of the final BI value to 0.70–0.79.

The location of LngREmph high values place and DifEntr high values areas in the ROI never overlaps [52]. Next, the LngREmph strongly indicates the sites in the consolidation region where a cancellous structure in the bone is absent. In contrary, DifEntr describes the sites with a scattered layout of bony trabeculae. Thus, entropy measures describe and detect good quality bone well [49,65], unlike LngREmph, which is a sensitive marker of bone atrophy in volume or quality, understood as fading of trabecularity. Therefore, both of these information were combined into one measure, which is the ratio of DifEntr to LngREmph (i.e., Bone Index). It is worth noting that the Bone Index is correlated to the age of the patient [52]. It should therefore monitor well the systemic influences on the bone-healing process [66]. This will be an interesting direction for future research.

As in the evaluation of a single feature from a co-occurrence matrix, the study of a composite BI feature whose components are derived from both the co-occurrence matrix (number of continuous long pixel chains of similar optical density) and the run-length matrix (entropy or chaotic nature of the trabecular structure measure) confirmed significant differences in the structure of the control regions of the mandibular head bone after osteosynthesis with magnesium screws (in comparison to the reference titanium fixations). Thus, it seems that this is not the result of feature selection (because three features from different measurement techniques are involved) but rather an altered internal structure of the entire mandibular head after magnesium osteosynthesis.

A different path of bone remodeling after magnesium osteosynthesis seems to also be confirmed by the SumOfSqr and Bone Index in the fracture line 12 months post-operation. Due to the fact that the screws still exist after 12 months for fixation in the mandibular head, the condition of the bone in the surrounding area of the resorbing magnesium screws should be observed. A probable factor influencing the quality of the union and the fate of the magnesium material is the physical characteristics of the alloy (except for the chemical composition) that determine the initial stiffness of the osteosynthesis. Fixation in the fragmented bone, allowing load transfer between the parts and promoting some interfragmentary movements for bone healing, may have positive meaning [67]. The interfragmentary movement is the relative movement between the bone fragments, which can appear during the patient’s occlusal loading. These micromovements in the fracture site are considered to promote bone callus growth and control of bone regeneration [68]. Hence, the controlled but possible masticatory activity of the patient, which is normal after ORIF contrary to conservative treatment [4], contributes to bone healing. These interfragmentary movements are crucial for the complex process of consolidating fracture [69]. Reasoning in this way, one can justify the favorable reduction in relative fixation stiffness by using magnesium screws (Young’s modulus 44 GPa, Table 3) in comparison with classical titanium screws having a higher elastic modulus. The micromovement may be desirable only in the first stage of bone healing. If the movement within the fracture does not decrease over time, no consolidation can be expected, which results in a nonunion [66].

The evaluation of the distribution of the intensity of BI values in the maps shown in Figure 3 indicates an interesting phenomenon. For both types of fixation, the condition of the articular surface is indicated as abnormal (low brightness of the central i.e., the most elevated part of the articular surface). This seems to be related to the observed remodelling of the mandibular heads in the area between the screws and the articular surface. This has been observed with titanium fixation [70,71]. Therefore, the same phenomenon seems to occur in magnesium osteosynthesis.

On the basis of digital image analysis of the fracture site treated with magnesium fixation, it should be noted that union of the fragments was achieved (consolidation was also achieved in the reference group). A feature of this union is a densification of the bone structure, especially after magnesium osteosynthesis. This is probably related to the method of magnesium alloy resorption by separating macroparticles and microparticles, which, until completely eliminated, is embedded in the trabecular bone [72,73,74]. A low Young’s modulus value should be considered too [54]. On one hand, the alloy enables the use of a material with mechanical properties similar to that of human compact bone. However, on the other hand, this alloy is not as stiff as the titanium one and may cause structural alterations in the post-fracture site during occlusal loading.

The longstanding utilization and commercialization of conventional bioinert implants still exist. However, biodegradable magnesium screws evince better performance than titanium and polymer implants [17,75]. This has led to commercialization of WE43 compressive screws—5000 orthopaedic surgeries have been performed in several countries. These implants were produced by subtractive CNC technology. However, 70% of surgeons prefer magnesium implants over polymeric ones. This indicates that magnesium screws have immense potential to replace the age-old bioinert and polymer-based implants. The success of magnesium implants is possible only because of intensive research conducted by surgeons and materials scientists. However, several mysteries remain regarding the degradation process of magnesium alloy, which is in the early stages of commercialization. Future research should focus on broadening the applications of magnesium alloys. Since there is no gold standard procedure to evaluate the biocompatibility of magnesium alloys, a comparison of their in vivo performance is difficult. Therefore, when a novel magnesium alloy is designed and fabricated, a head-to-head comparison with the performance of known resorbable alloys is vital. For this purpose, the size of the implant, location, and animal model of the novel magnesium alloys should be the same as that of the reference (for instance, WE43 alloy). In order to verify the safety of magnesium implants and their applications in non-load-bearing regions, research on small animals is sufficient. However, in order to mimic load-bearing conditions similar to that of the condylar head in humans, large animal models are needed. The performance of magnesium implants is comparable to that of titanium to treat fractures in the upper limb and elbow regions and some parts of the lower limb i.e., foot and ankle region [75]. Hence, future studies should focus on extending their application to other surgical areas like the *processus condylaris mandiblae*.

Additive technology is another emerging production method for magnesium alloy medical implants. There is currently a growing interest in three-dimensional additive technologies based on metal raw materials. This is undoubtedly related to the advanced adaptation of the physical and chemical properties of metal alloys and powders to specific applications. This results in excellent final results. In the condylar head region, custom repositioning plates/mesh according to the Ukrainian concept is a highly promising prospect [76,77]. Full adaptation to the details of the mandibular head anatomy can be cited as specific applications [78] to induce faster cell growth, more cell divisions, and eventually better quality bone consolidation. [79]. The fusing and melting of metal powders into complex anatomical shapes were not achieved by older traditional subtractive techniques and the literature indicates that such complex geometries were a barrier in respect to classical manufacturing techniques [80]. Three-dimensional printing effectively expands manufacturing capabilities while simultaneously simplifying production and reducing the cost of creating personalized implants, screws, plates, templates, and replacements for lost anatomical elements [81]. All of this converges positively in medical components that can be constructed from magnesium alloys.

The limitations of this study are the small number of included patients and a follow-up period shorter than the time of screw resorption. There were also demographic (in terms of internal diseases) and epidemiological (in terms of the prevalence of bilateral fractures) differences in the presented groups. However, a strong point of this study is the objective assessment of the union characteristics of the bone fragments fixed by magnesium implants.

## 5. Conclusions

Radiological analysis, supported by digital texture analysis, indicates that osteosynthesis performed with compressible, headless screws made of WE43 magnesium alloy produces stable long-term results. The union achieved in rigid internal fixation of the mandibular head is qualitatively similar with magnesium alloy to that achieved with screws made of titanium alloy. However, the resorption process of the screws was found to be incomplete after 12 months, evidenced by a marked densification of the bone structure at the fracture site.

The limitation of this analytical method is the quality of the imaging results and the acquiring technique. The amount of information that can be extracted using mathematical methods depends on the source images. It will be interesting to check the quality of the bone union induced by magnesium fixation on micro-CT and high tesla magnetic resonance imaging.

## Figures and Tables

**Figure 1 materials-15-02230-f001:**
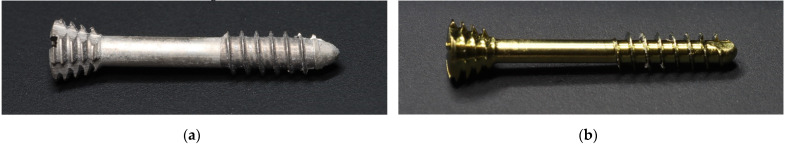
General appearance of the compressive headless screws used in the study. These screws are ChM’s dedicated designs for mandibular head fixation. (**a**) magnesium alloy 2.2 mm screw (test group); (**b**) titanium alloy 1.7 mm screw (reference group). Compressibility is ensured by different thread pitch. The large pitch at the tip of the screw ensures a large longitudinal movement with every turn in proximal bone fragments. On the other hand, a small thread pitch at the socket end leads to a small longitudinal movement of the screw in distal bone fragments, which ensures a compression effect in the fracture gap.

**Figure 2 materials-15-02230-f002:**
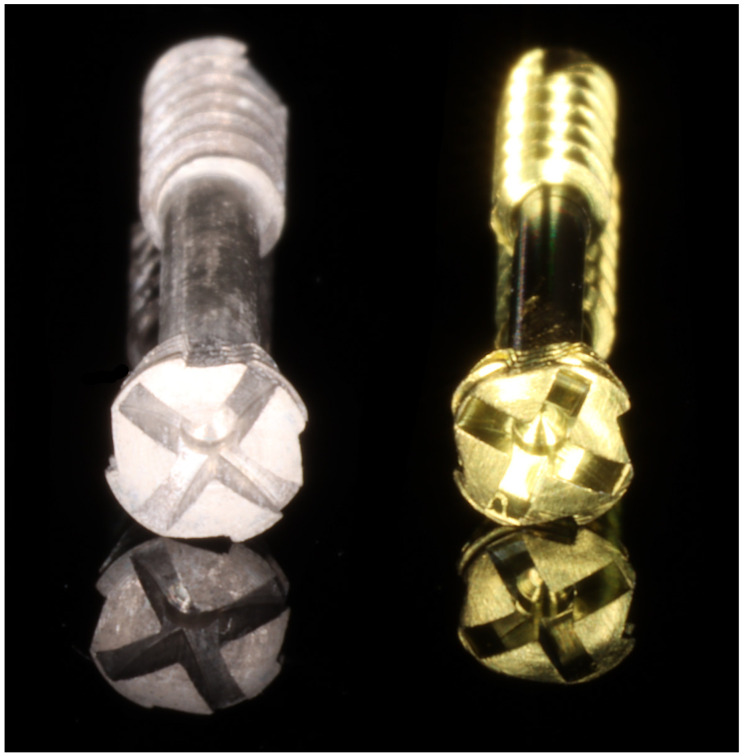
The compression screws used have cross sockets and peripheral notches to collect bone chips. It is assumed that both screw types will be immersed in the bone in their entirety.

**Figure 3 materials-15-02230-f003:**
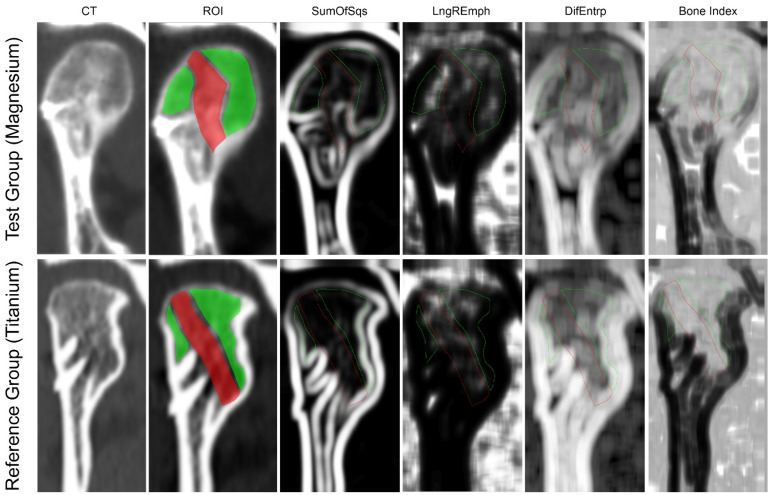
Examples of bone union after 12 months from osteosynthesis in the Magnesium and Titanium group observed by computed tomography (CT) mapping of regions of interest for texture analysis (ROI red: post-fracture site; green: control cancellous bone in the mandible head). The results of calculating the sum of squares of optical densities over a distance of two pixels (SumOfSqr), the frequency of long chains of pixels of similar optical density (LngREmph), the differential entropy in the image (DifEntrp), and the Bone Index were mapped.

**Table 1 materials-15-02230-t001:** Data describing the test group versus the reference group.

Variable	Test Group (Magnesium)	Reference Group (Titanium)	Between-Group Difference
Age	30 ± 20 years	35 ± 15 years	*p* = 0.125
Gender (female, male)	13%, 39%	8%, 39%	*p* = 0.460
Tabaco smokers	11%	0%	*p* = 0.087
Co-morbidity (internal)	11%	20%	*p* < 0.05
Taking systemic medication	6%	0%	*p* = 0.383
Reason of injury (RTA ^1^, assault, fall)	15%, 26%, 28%	17%, 7%, 7%	*p* = 0.063
Type of fracture (B, C)	17%, 43%	4%, 37%	*p* = 0.173
Mandible ramus shortening (M, P) ^2^	13%, 46%	0%, 41%	*p* = 0.053
Location (single, bilateral fracture)	41%, 19%	13%, 28%	*p* < 0.05

^1^ Road Traffic Accident. ^2^ M—fracture without ramus shortening, P—fracture with ramus shortening.

**Table 2 materials-15-02230-t002:** Comparison of the materials in the screws in the test and reference groups.

Magnesium Alloy MgYREZr	Composition	Titanium Alloy Ti6Al7Nb	Composition
Magnesium	Balance	Titanium	Balance
Yttrium	3.5–4.5%	Aluminium	6.2950%
Rare Earth	2.5–3.5%	Niobium	6.8700%
Zirconium	<0.6%	Tantalum	<0.001%
Iron	<80 ppm	Iron	0.175%
Manganese	<200 ppm	Oxygen	0.166%
Aluminium	<200 ppm	Carbon	0.0065%
Silicon	<100 ppm	Nitrogen	0.003%
Copper	<100 ppm	Hydrogen	0.0024%
Nickel	<30 ppm	Nickel	0.0205%
Beryllium	<20 ppm	Vanadium	0.0195%
		Other Single Trace	<0.05%
		Total Trace Elements	0.104%

**Table 3 materials-15-02230-t003:** Physical properties of the materials used in the experiment (see Table 2).

Property	Magnesium Alloy MgYREZr	Titanium Alloy Ti6Al7Nb
Tensile strength Rm	Min. 280 MPa	Min. 988 MPa
Yield strength R_p0.2_	Min. 200 MPa	Min. 800 MPa
Elongation ε	Min. 10%	Min. 10%
Density	1.84 g/cm^3^	4.52 g/cm^3^
Young modulus	44 GPa	103 GPa
Poisson’s ratio	0.27	0.33
Melting range	540–640 °C	1530–1590 °C

**Table 4 materials-15-02230-t004:** Comparison of the fracture site 12 months post-operation by means of texture analysis with SumOfSqr and Bone Index features.

Measured Feature and Site i.e., ROI	Test Group (Magnesium)	Reference Group (Titanium)	Between-Group Difference
Sum of Squares in the fracture line	109 ± 5 ^1^	10 6 ± 6 ^1^	*p* < 0.05
Sum of Squares in cancellous control bone	99 ± 7	94 ± 7	*p* < 0.05
Bone Index in the fracture line	0.70 ± 0.25 ^2^	0.79 ± 0.23 ^2^	*p* = 0.153
Bone Index in the cancellous control bone	0.97 ± 0.23	1.08 ± 0.07	*p* < 0.05
Sum of Squares Relative	1.11 ± 0.10 ^3^	1.13 ± 0.11 ^3^	*p* = 0.374
Bone Index Relative	0.73 ± 0.25 ^4^	0.72 ± 0.19 ^4^	*p* = 0.975

^1^ Observed opaque islands in healed fracture site texture generated an increase in the value of SumOfSqrt. ^2^ The low value is due to the low diversity within the radio-opaque islands, detected as extensive areas of uniformly elevated optical density. ^3^ In both groups, the relative values increased, confirming that SumOfSqr was higher in the remodeled fracture line with both magnesium and titanium osteosynthesis. ^4^ In both groups, the BI value decreased similarly in the osteosynthesis site.

## Data Availability

The data presented in this study are available on request from the corresponding author. The data are not publicly available due to an ongoing multicentre project.

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
