# Peer review of "Bone Union Quality after Fracture Fixation of Mandibular Head with Compression Magnesium Screws"

_materials, 2022, doi:10.3390/ma15062230_

Round 1

Reviewer 1 Report

This study described a clinical research on using compression Mg alloy screws in the fracture fixation of mandibular head to investigate the bone union. The authors have a clear objective and the result also verified their hypothesis. Although the manuscript was reasonably written as a clinical study, the reviewer has a concern of the suitability of the study to be published in "Materials" journal. It is suggested the authors may resubmit the manuscript to a clinical research related journal and it may significantly enhance its visibility. As a research article regarding the biomaterial topic, the reviewer suggests the authors to add or revise the manuscript in terms of points as shown below.

  1. The introduction should add a background description on Mg and other metal alloys that have been utilized in mandibular head and other related bone fracture union, focusing on the material advantages and disadvantages in these application cases.
  2. Although this is a real clinical study, the characterization of materials other than the observation of bone union should be performed as well. For example, the material degradation rate or re-absorbable rate over the 12 month bone union period may be evaluated under certain microscope. The metal residue in the patient body should also be evaluated.
  3. The Mg and Ti alloys should be compared before they were implanted, such as their mechanical or physical properties, because these may affect their performance during the period of bone union.
  4. The manuscript should be revised toward the effect of Mg alloy screws on the bone union. The introduction, materials and methods, results and discussion should emphasize such a topic.   

Author Response

Comments and Suggestions for Authors
This study described a clinical research on using compression Mg alloy screws in the fracture fixation of mandibular head to investigate the bone union. The authors have a clear objective and the result also verified their hypothesis. Although the manuscript was reasonably written as a clinical study, the reviewer has a concern of the suitability of the study to be published in "Materials" journal. It is suggested the authors may resubmit the manuscript to a clinical research related journal and it may significantly enhance its visibility. As a research article regarding the biomaterial topic, the reviewer suggests the authors to add or revise the manuscript in terms of points as shown below.

The introduction should add a background description on Mg and other metal alloys that have been utilized in mandibular head and other related bone fracture union, focusing on the material advantages and disadvantages in these application cases. 
ANSWER: It has been added.

Although this is a real clinical study, the characterization of materials other than the observation of bone union should be performed as well. For example, the material degradation rate or re-absorbable rate over the 12 month bone union period may be evaluated under certain microscope. The metal residue in the patient body should also be evaluated.
ANSWER: The rate of resorption was taken into account in a previous study (Kozakiewicz, M. Change in Pull-Out Force during Resorption of Magnesium Compression Screws for Osteosynthesis of Mandibular Condylar Fractures. Materials 2021, 14, 237. doi:10.3390/ma14020237) so it is cited here at the request of the reviewer. As for micro- or macro-particles of magnesium or titanium, this would undoubtedly be interesting and important, but the bioethics committee's approval for post-treatment patient tissue sampling was not obtained. However, from the publications already known, many of the effects of resorbability are positive [Li, M.; Ren, L.; Li, L.H.; He, P.; Lan, G.B.; Zhang, Y.; Yang, K. Cytotoxic Effect on Osteosarcoma MG-63 Cells by Degradation of Magnesium, J Mater Sci Tech, 2014, 30, 888-893, doi:10.1016/j.jmst.2014.04.010. Qu, X.H., Jin, F.C., Hao, Y.Q., Zhu, Z.A., Li, H.W., Tang, T.T., Dai, K.R. Nonlinear association between magnesium intake and the risk of colorectal cancer. Eur. J. Gastroen. Hepat. 2013, 25, 309–318. Wang, Q., Jin, S., Lin, X., Zhang, Y., Ren, L., Yang, K. Cytotoxic effects of biode gradation of pure Mg and MAO-Mg on tumor cells of MG63 and KB. J. Mater. Sci. Tech, 2014, 30, 487-492, doi:10.1016/j.jmst.2014.03.004, Mazur, A.; Maier, J.A.; Rock, E.; Gueux, E.; Nowacki, W.; Rayssiguier, Y. Magnesium and the inflammatory response: potential physiopathological implications. Arch. Biochem. Biophys. 2007, 458, 48–56. Peng, Q.M.; Li, K.; Han, Z.S.; Wang, E.D.; Xu, Z.G.; Liu, R.P.; Tian, Y.J. Degradable magnesium-based implant materials with anti-inflammatory activity. J. Biomed. Mater. Res. 2013, A 101,1898–1906. Li, Y.; Liu, G.W.; Zhai, Z.J.; Liu, L.N.; Li, H.W.; Yang, K.; Tan, L.L.; Wan, P.; Liu, X.Q.; Ouyang, Z.X.; Yu, Z.F.; Tang, T.T.; Zhu, Z.N.; Qu, X.H.; Dai, K.R. Antibacterial properties of magnesium in vitro and in an in vivo model of implant-associated methicillin-resistant staphylococcus aureus infection. Antimicrob. Agents Ch. 2014, 58, 7586–7591. Ren, L.; Lin, X.; Tan, L.L.; Yang, K. Effect of surface coating on antibacterial behawior of magnesium based metals. Mater. Lett. 2011, 65, 3509–3511. Robinson, D.A.; Griffith, R.W.; Shechtman, D.; Evans, R.B.; Conzemius, M.G. In vitro antibacterial properties of magnesium metal against Escherichia coli, Pseudomonas aeruginosa and Staphylococcus aureus. Acta Biomater. 2010, 6, 1869–1877. Chen, Y.J.; Xu, Z.G.; Smith, C.; Sankar, J. Recent advances on the development of magnesium alloys for biodegradable implants. Acta Biomater. 2014, 10, 4561–4573. Liu, C.; Fu, X.K.; Pan, H.B.; Wan, P.; Wang, L.; Tan, L.L.; Wang, K.H.; Zhao, Y.; Yang, K.; Chu, P.K. Biodegradable Mg-Cu alloys with enhanced osteogenesis, angiogenesis, and long-lasting antibacterial effects. Sci. Rep.-UK 2016, 6, 27374. Zhai, Z.J.; Qu, X.H.; Li, H.W.; Yang, K.; Wan, P.; Tan, L.L.; Ouyang, Z.X.; Liu, X.Q.; Tian, B.; Xiao, F.; Wang W.G.; Jiang, C., Tang, T.T.; Fan, Q.M.; Qin, A.; Dai, K.R. The effect of metallic magnesium degradation products on osteoclast-induced osteolysis and attenuation of NF-kappa B and NFATc1 signaling. Biomaterials 2014, 35, 6299–6310]. These publications have been included in the manuscript.

The Mg and Ti alloys should be compared before they were implanted, such as their mechanical or physical properties, because these may affect their performance during the period of bone union.
ANSWER: Some of that properties were tested and presented in papers which are cited in manuscript [e.g. Kozakiewicz, M. Are Magnesium Screws Proper for Mandibular Condyle Head Osteosynthesis? Materials 2020, 13, 2641. Doi:10.3390/ma13112641. Kozakiewicz, M. Change in Pull-Out Force during Resorption of Magnesium Compression Screws for Osteosynthesis of Mandibular Condylar Fractures. Materials 2021, 14, 237. doi:10.3390/ma14020237]. It has been additionally included in tables in Material & Method section.

The manuscript should be revised toward the effect of Mg alloy screws on the bone union. The introduction, materials and methods, results and discussion should emphasize such a topic.  
ANSWER: it has been emphased.

Reviewer 2 Report

- Improve the introduction with data about materials used for Fracture Fixation of Mandibular Head from the literature.

- Add the chemical composition (EDS) for titanium and magnesium screws.

- Highlight better which is the novelty of the work?

- What is the status of the literature according to your work? Make a comparison between the results obtained by you and another previous research.

- Add more conclusions. Complete the conclusions with the limitations of the proposed methodology. Also write future research.

- Generally, the quality of the writing could be improved.

Author Response

Comments and Suggestions for Authors
- Improve the introduction with data about materials used for Fracture Fixation of Mandibular Head from the literature.
ANSWER: It has been added.

- Add the chemical composition (EDS) for titanium and magnesium screws.
ANSWER: It has been added.

- Highlight better which is the novelty of the work?
ANSWER: The background and novelty of performed study has been developed and indicated in the Introduction and Discussion sections.

- What is the status of the literature according to your work? Make a comparison between the results obtained by you and another previous research.
ANSWER: According to my knowledge it is the first publication supported in statistics for magnesium mandible condylar head osteosynthesis. Only previous literature in that field consist of two paperes:
1. one case presented: Leonhardt, H.; Franke, A.; McLeod, N.M.H.; Lauer, G.; Nowak, A. Fixation of fractures of the condylar head of the mandible with a new magnesium-alloy biodegradable cannulated headless bone screw. Br J Oral Maxillofac Surg 2017, 55, 623-625. Doi: 10.1016/j.bjoms.2017.04.007
2. 6 case presented [incluning that one from teh first case report]: Leonhardt, H.; Ziegler, A.; Lauer, G.; Franke, A. Osteosynthesis of the mandibular condyle with magnesium-based biodegradable headless compression screws show good clinical results during a 1-year follow-up period. J Oral Maxillofac Surg. 2021, 79, 637-643. doi: 10.1016/j.joms.2020.02.025
The current situation is that there are no statistical references in the literature with which to compare my findings.

- Add more conclusions. Complete the conclusions with the limitations of the proposed methodology. Also write future research.
ANWER: It has been added.

- Generally, the quality of the writing could be improved.
ANSWER: It has been improved.

Reviewer 3 Report

The authors have done a radiological evaluation of the quality of bone union after 57 osteosyntheses of a mandibular head fracture with compression magnesium screws. Although this is a well-written manuscript, there are a few points that can improve the paper. Introduction: You cited your previous article (Ref#18). However, there is no connection made between these investigations. What are the advantages of the present one? How about other similar studies? Materials and Methods: The demographic information of subjects are not provided. Is there any specific confounding factor that can affect your results? Have you used randomization? what was your technique? References: They should be rechecked. There are a few errors such as Ref: 6, 8, 19, 25, 28, and 33.

Author Response

Comments and Suggestions for Authors
The authors have done a radiological evaluation of the quality of bone union after 57 osteosyntheses of a mandibular head fracture with compression magnesium screws. Although this is a well-written manuscript, there are a few points that can improve the paper. Introduction: You cited your previous article (Ref#18). However, there is no connection made between these investigations. What are the advantages of the present one?
ANSWER: I dare to point out that in the paper originally published as paper number 18 [Kozakiewicz, M.; Gabryelczak, I.; Bielecki-Kowalski, B. Clinical Evaluation of Magnesium Alloy Osteosynthesis in the Mandibular Head. Materials 2022, 15, 711. Doi.org/10.3390/ma15030711] There is a first application of qualitative assessment of the quality of the adhesion (see Fig.4 and Table 1). And this is the title of this paper. The authors feel that paper #18 was the introduction and first touches on the topic of investigating the quality of adhesion in magnesium fixations. And in the present work it is considerably developed.

How about other similar studies?
ANSWER: According to my knowledge it is the first publication supported in statistics for magnesium mandible condylar head osteosynthesis. Only previous literature in that field consist of two paperes:
1. one case presented: Leonhardt, H.; Franke, A.; McLeod, N.M.H.; Lauer, G.; Nowak, A. Fixation of fractures of the condylar head of the mandible with a new magnesium-alloy biodegradable cannulated headless bone screw. Br J Oral Maxillofac Surg 2017, 55, 623-625. Doi: 10.1016/j.bjoms.2017.04.007
2. 6 case presented [incluning that one from teh first case report]: Leonhardt, H.; Ziegler, A.; Lauer, G.; Franke, A. Osteosynthesis of the mandibular condyle with magnesium-based biodegradable headless compression screws show good clinical results during a 1-year follow-up period. J Oral Maxillofac Surg. 2021, 79, 637-643. doi: 10.1016/j.joms.2020.02.025
The current situation is that there are no statistical references in the literature with which to compare my findings.

Materials and Methods: The demographic information of subjects are not provided.
ANSWER: Demographic results from patients analysis were add to the Material & Method section.

Is there any specific confounding factor that can affect your results?
ANSWER: Possible systemic (age) on bone healing which can be monitoring by Bone Index was mentioned in previous Discusion. The group presented, although the largest currently published, is relatively small. This requires further research to increase the size of the groups. Ideally this should be done on a multicentre basis. In the reference group, there are more patients with concomitant systemic diseases (although the two groups do not differ in the drugs they take). In addition, there were more single fractures in the group treated with magnesium alloy, while the group treated with titanium alloy had more odustronic fractures. This was added in Materials and Methods and at the end of the Discussion.

Have you used randomization? what was your technique?
ANSWER: Yes, randomisation was used in the study. Patients constituting the clinical material were operated on from Tuesday to Friday each week. 
Patients operated on Tuesdays and Thursdays received titanium fixation, while patients operated on Wednesdays and Fridays received magnesium fixation. The choice of fixation material was decided on the day the patient arrived at the hospital. 

References: They should be rechecked. There are a few errors such as Ref: 6, 8, 19, 25, 28, and 33.
ANSWER: The correctness of the citation and the way in which the references were written has been checked. Position 19 has been removed. Some DOI numbers have been added. All referenced has been rebuilt.

Round 2

Reviewer 1 Report

NA

Reviewer 2 Report

The paper has been significantly improved, it can be accepted for publication